# Immunomodulatory Role of *Rosmarinus officinalis* L., *Mentha* x *piperita* L., and *Lavandula angustifolia* L. Essential Oils in Sheep Peripheral Blood Mononuclear Cells

**DOI:** 10.3390/vetsci11040157

**Published:** 2024-04-01

**Authors:** Maria Giovanna Ciliberti, Marzia Albenzio, Agostino Sevi, Laura Frabboni, Rosaria Marino, Mariangela Caroprese

**Affiliations:** Department of Agriculture, Food, Natural Resources, and Engineering (DAFNE), University of Foggia, 71122 Foggia, Italy; marzia.albenzio@unifg.it (M.A.); agostino.sevi@unifg.it (A.S.); laura.frabboni@unifg.it (L.F.); rosaria.marino@unifg.it (R.M.); mariangela.caroprese@unifg.it (M.C.)

**Keywords:** sustainability, immune response, cytokines, one health

## Abstract

**Simple Summary:**

The increasing demand for livestock production with a more sustainable approach to reduce greenhouse emissions offers the opportunity to test essential oils (EOs), as natural treatments, for their effect on rumen activity, as well as the use of antibiotics, for their antimicrobial activities. However, little information is available on the effects of EOs on the proliferative immune response and cytokine production. Therefore, the present paper is aimed at evaluating the effect of the *Mentha* x *piperita* L., *Rosmarinus officinalis* L., and *Lavandula angustifolia* L. EOs on sheep peripheral blood monocular cells’ bio-response in terms of viability, proliferation, and cytokine secretion. The main results obtained encourage the implementation of these EOs as feed additives, in in vivo studies, to improve the animals’ immune competence, especially those under specific physiological or environmental stressors.

**Abstract:**

Recently, the uses of essential oils (EOs) as rumen modifiers, anti-inflammatory agents, and antioxidants were demonstrated in livestock. In the present study, the role of *Mentha* x *piperita* L. (MEO), *Rosmarinus officinalis* L. (REO), and *Lavandula angustifolia* L. (LEO) EOs in an in vitro sheep model of inflammation was investigated. With this aim, peripheral blood mononuclear cells (PBMCs) were treated with incremental concentrations (3, 5, 7, and 10%) of each EO to test their effects on cell viability and proliferation and on interleukin (IL)-6, IL-10, and IL-8 secretion. The PBMCs were stimulated by Concanavalin A (ConA) alone or in combination with lipopolysaccharide (LPS) mitogen. The positive and negative controls were represented by PBMCs in the presence or absence, respectively, of mitogens only. The cell viability and proliferation were determined by XTT and BrdU assays, while the cytokines were analyzed by ELISA. The EO treatments did not affect the viability; on the contrary, the PBMC proliferation increased in presence of all the EOs tested, according to the different percentages and mitogens used. The IL-10 secretion was higher in both the REO and the LEO tested at 3% than in the positive control; furthermore, the IL-8 level was influenced differently by the various EOs. The present data demonstrate that EOs may modulate the immune response activated by inflammation.

## 1. Introduction

Essential oils (EOs) are classified as a group of organic compounds originating from the secondary metabolism of plants, which exhibit antimicrobial activity to control diseases [1]. Chemically, EOs are a mixture of different substances, including about 20 to 60 components, including terpenoids, alcohols, aldehydes, hydrocarbons, ketones, esters, and ethers [2]. The most representative molecules, accounting for 90% of the total composition, are the monoterpenes (limonene, thymol, carvacrol, linalool, carvone, geranyl acetate), followed by the phenylpropanoids, derived from phenylalanine [3]. Recently, these bioactive compounds have attracted interest in relation to animal feed for their ascertained ability to modify the rumen and gut functioning, contributing to the reduction in global greenhouse gas emissions [4] and, therefore, to the increase in the sustainability of livestock productions. Furthermore, in light of the “One World—One Health” concept, it could be crucial create innovative strategies to control the emergence of infections [5] and to reduce antimicrobial resistance through the exploitation of EOs as a safe potential replacement for therapy with antibiotics [6]. Indeed, recent evidence has demonstrated that EOs might modulate the immune system, antioxidant activity, and udder health of dairy animals [7]. Moreover, EOs have been found to improve rumen fermentation efficiency by altering gut microbiota and sustaining the integrity of the intestinal barrier, gut immune responses, and growth performance, supporting small-ruminant-derived products [8]. 

Despite the increasing research on EOs, very few reports are available on the specific effects of EOs on immune responses with a particular focus on inflammatory processes. It has been suggested that the bioactive molecules from EOs could target multiple inflammatory signaling pathways, thus masking the possible identification of a leader anti-inflammatory molecule [9]. Hence, in veterinary science, it might be essential to explore the anti-inflammatory effects of EOs in order to increase the knowledge on the use of aromatic plants as anti-inflammatory drugs for the treatment of inflammation [10]. In the study of ruminants, the evaluation of immune competence can be determined throughout the blastogenic response of peripheral blood mononuclear cells (PBMCs) to liposaccharide mitogen (LPS) [11]. Indeed, the PBMCs of dairy cows have shown low responsiveness to LPS stimulation after calving, suggesting an immune-depression state around this time [11]. In addition, LPS is a mitogen derived from microorganisms able to initiate the proliferation of B-cells, which can be used in combination with other mitogens derived from plants, including concanavalin A, a type of T-cell mitogen [12], in order to optimize the lymphocyte proliferation rate in both in vivo and in vitro trials [13]. 

Furthermore, studies have underlined the key mediating role of cytokines secreted by immune cells, including PBMC, in the coordination of the physiological response to stress stimuli in both health or disease states [14,15] by regulating the activation, replication, chemotaxis, and apoptosis of immune cells [16]. Our hypothesis was that EOs could have an immunomodulatory role, by sustaining the immune responses of small ruminants, in terms of cell proliferation and cytokine production with pro- and anti-inflammatory actions.

Therefore, in this study, we investigated the effects of *Mentha* x *piperita* L., *Rosmarinus officinalis* L., and *Lavandula angustifolia* L. oils (EOs) on sheep PBMC viability, proliferation, and cytokine secretion, in the presence of ConA and the combination of ConA and LPS stimulation.

## 2. Materials and Methods

### 2.1. Plant Sample Collection and Extraction of Essential Oils

*Mentha* x *piperita* L. var. *citrata* (Ehrh.) Briq, *Rosmarinus officinalis* var. *Pyramidalis* L., and *Lavandula angustifolia* var. *Royal purple* L. were cultivated in an open field on the “Bonomelli” farm. The soil underwent plowing to a depth of 35 cm and one treatment with a disc harrow at a depth of 20 cm. Each crop was transplanted into continuous rows spaced at 1 × 1 m, and standard agronomic practices for aromatic crops (fertilization, drip irrigation, weed control, etc.) were developed. Fresh collected (at 11 a.m.) leaves and flowers of *M.* x *piperita, R. officinalis*, and *L. angustifolia* (only flowers) were used for hydrodistillation using Clevengertype apparatus [17] in order to extract EOs. The extraction lasted two hours for each plant. One kilogram of fresh plant (leaves and flowers) was submitted to hydrodistillation with 7 L of distilled water (according to the European Pharmacopoeia). At the end of the extraction, the accumulated EOs floated on the aqueous phase (hydrolat). All the material was transferred into a glass separating funnel for four hours to obtain a total separation (EOs/hydrolate). The EOs were stored in hermetically sealed dark-glass containers and kept in rooms at 5–6 °C. The determination of the oil compositions was performed by gas chromatography (GC) coupled with mass spectrometry (MS) analysis according to Adams’ method [18]. 

### 2.2. Animals and Experimental Treatments

Dairy sheep enrolled in this study were from a commercial farm located in Foggia (Italy). The guidelines of the EU Directive 2010/63/EU (2010) on the protection of animals used for experimental and other scientific purposes were followed. The animal study protocol was approved by the Ethics Committee of the University of Foggia (protocol number 0002302). Veterinarians examined all the animals to exclude the presence of any signs of disease. Sheep were chosen with a random design, and an in vitro experiment was performed, with four incremental percentages (3, 5, 7, and 10%) of rosemary EO (REO), mint EO (MEO), and lavender EO (LEO), respectively. For the biological experiments, stock solutions of each EO were prepared by solubilizing in dimethyl sulfoxide (DMSO,1:1). Next, several dilutions were prepared in a cell culture medium and evaluated for their in vitro proliferative and anti- or pro-inflammatory activity, measured by cytokine secretion. All the samples were protected from light at −20 °C until use. The final concentration of DMSO in cell culture was less than 0.5% for all EOs tested to exclude any toxic effect on cells, as previously reported by Ciliberti et al. [19].

### 2.3. Isolation of Sheep Peripheral Blood Mononuclear Cells

A density-gradient centrifugation method was applied to isolate PBMCs according by Ciliberti et al.’s modification [20] of Wattegedera et al.’s method [21]. Briefly, Na-heparinized blood samples (15 mL) were collected from the jugular veins of sheep, diluted in cold PBS (1:1, pH 7.4), and centrifuged at 670× *g* at 4 °C for 20 min. Buffy coat containing leukocytes was recovered, layered over 10 mL of Hystopaque gradient (Merk/Sigma-Aldrich, Darmstadt, Germany, 1.077 g/mL), and centrifuged at 1130× *g* at 25 °C for 30 min. The mononuclear cell ring was recovered, washed with prewarmed (37 °C) Hanks’ Balanced Salt solution (HBSS), and finally suspended at a final concentration of 2 × 10^6^ cells/mL in Iscove’s modified Dulbecco’s medium (Merk/Sigma-Aldrich, Darmstadt, Germany) supplemented with 10% fetal bovine serum (Merk/Sigma-Aldrich, Darmstadt, Germany) and 50 μg/mL gentamicin (Merk/Sigma-Aldrich, Darmstadt, Germany). After the isolation step, the PBMC viability was determined by trypan blue dye exclusion using Burker chamber, reaching >96%.

### 2.4. XTT Cell Proliferation Assay

The evaluation of in vitro cell viability was performed by using TACS^®^ XTT Cell Proliferation Assay (R&D Systems, Inc., Minneapolis, MN, USA). This assay was based on the evaluation of the metabolically active cells through the direct cleavage of the tetrazolium salt (yellow dye) to formazan (orange dye,) initiated by the succinate–tetrazolium reductase system of the mitochondria. The protocol was carried out according to the manufacturer’s suggestions. Briefly, 100 μL of cell suspension (1 × 10^5^ cells/mL) was seeded into quadruplicate in U-96-well plate. The absorbance was set at 450 nm in a microplate reader. Data were presented as the percentage change in absorbance relative to the experimental control (negative control, CNS).

### 2.5. Bromodeoxyuridine Proliferation Assay

The proliferative response of PBMCs to EO treatment was measured by bromodeoxyuridine (BrDU) incorporation during DNA synthesis. In brief, a PBMC suspension measuring 100 µL (1 × 10^6^ cells/mL) was seeded into quadruplicate in a U 96-well plate. Cells were stimulated by the addition of a mixture of concanavalin A (ConA, Sigma-Aldrich), at a final concentration of 5 μg/mL, and lipopolysaccharide (LPS, Sigma-Aldrich), at final concentration of 1 μg/mL, as previously reported by Ciliberti et al. [22]. Moreover, stimulation with only concanavalin A (ConA, 5 μg/mL, final concentration) was performed. Both the PBMCs stimulated with ConA and LPS and those stimulated with ConA only were treated with an incremental concentration (3, 5, 7, and 10%) of REO, MEO, and LEO. In the in vitro trial, the positive controls were represented by PBMCs stimulated with Con A and LPS (CS_ConALPS), with Con A only (CS_ConA), or with LPS only (CS_LPS). Moreover, the negative control (CNS) was characterized by PBMCs in presence of culture medium only and with no addition of DMSO. The plates were cultured in a humidified incubator under 5% CO_2_ at 37 °C for 24 h. After incubation time, cell-free supernatants were collected, after a centrifugation step at 300× *g* for 10 min, and stored at −20 °C until ELISA to measure cytokine secretion. Next, the PBMCs remaining in the bottoms of wells were incubated with BrdU (Exalpha Biologicals Inc., Shirley, MA, USA) to test the proliferative response to an in vitro EOs stimulation. After 18 h of incubation, the incorporation of BrdU during DNA synthesis was measured by determining optical density at 450 nm by using a spectrophotometer (PowerWave XS, Biotek, Swindon, UK).

### 2.6. Determination of IL-6, IL-10, and IL-8 in PBMC Supernatant

The secretion of IL-6, IL-10, and IL-8 cytokines in PBMC supernatant was determined in duplicates by a sandwich ELISA according to methods reported by Ciliberti et al. [20,22]. Plates were coated overnight at 4 °C with the capture antibodies, represented by mouse monoclonal antibody (mAb) anti-sheep IL-6, (Clone 4B6, Biorad Ltd., Hercules, CA, USA), anti-bovine IL-10 (Clone CC318, Biorad Ltd., Hercules, CA, USA), and anti-bovine IFN-γ (Clone CC330, Biorad Ltd., Hercules, CA, USA). As detecting antibodies, the rabbit polyclonal anti-ovine IL-6, the biotinylated mouse anti-bovine IL-10 mAb (Clone CC320, Biorad Ltd., Hercules, CA, USA), and the biotinylated anti-bovine IFN-γ antibody (clone CC302, Biorad Ltd., Hercules, CA, USA) were added, respectively. All the bovine antibodies involved in the sandwich ELISA demonstrated cross reactivity with ovine species. In each assay a standard curve was built using scalar dilution of the recombinant ovine IL-6 protein (Cusabio Biotech Co., Wuhan, China), recombinant bovine IL-10 (Biorad Ltd., Hercules, CA, USA), and recombinant bovine IFN-γ (Kingfisher Biotech, St Paul, MN, USA). The reading was set at 450 nm (Power Wave XS, Biotek, Charlotte, VT, USA). Data were expressed in nanograms of IL-6, IL-10, and IL-8 per milliliter.

### 2.7. Statistical Analysis

Results were presented as the means ± the standard error of the mean. The Shapiro–Wilk test was used for analysis of the data distribution. After confirmation of normal data, the comparisons between cell culture treatments were made using one-way analysis of variance (ANOVA) by using GraphPad Prism 5 software (GraphPad Software, La Jolla, CA, USA). The Tukey post hoc adjustment for multiple comparisons was used to set the differences between mean cell culture treatments. Statistical differences of *p* < 0.05 were considered significant. The secretion of IL-6, IL-10, and IL-8 cytokines in presence of Con A stimulation and EOs tested was below of the limit of detection; therefore, for statistical analysis, only the secretion of cytokines of PBMC supernatant from the stimulation in presence of both ConA and LPS mitogens was analyzed.

## 3. Results and Discussion

### 3.1. Essential Oils Composition

Table 1 shows the results of the GC–MS analysis of the EOs employed for the PBMC treatment. Twenty-four compounds were identified in the EO of *R. officinalis* (REO); the main constituents were 1,8-cineole (25.2%), α-pinene (20%), camphor (16.2%), and camphene (10.8%), accounting for a 72.2% of the total compounds identified. In a study by Gachkar et al. [23], lower percentages of the major compounds α-pinene and 1,8-cineole were found, registering values of 14.9%, and of 7.43%, respectively. Previous studies on REO composition reported slight differences between the main compounds [24,25,26]; these results can be ascribed to the effect of the climate on the relevant plants, which, in turn, are able to modify their chemical composition [23]. Moreover, our findings were consistent with a prior observation by Akrout et al. [27], who identified three distinct *R. officinalis* chemotypes that contained EOs with equal amounts (20–30%) of 1,8-cineole, α-pinene, and camphor from Morocco, Tunisia, Turkey, Greece, Yugoslavia, Italy, France, and Algeria. A total of 16 compounds were identified in the EO of *M. piperita* L. (MEO), characterized by a huge amount of piperitenone oxide (51.25%) and a lower amount of 1,8-cineole (17.4%, eucalyptol) and limonene (11.8%), out of a total of 96.65% compounds identified. In a study by Gracindo et al. [28], some genotypes of Mint (*Mentha* spp.) were characterized by EOs with a high content piperitone oxide, including *Mentha suavelons* (79.0%) and *Mentha spicata* (65.5%). Accordingly, the chemical composition of mint genotypes was related to the region of origin, showing different percentages of oxides, particularly piperitone oxide and piperitenone oxide, the major specimens [29,30,31,32]. Finally, the chemical composition of *L. angustifolia* essential oil (LEO) was characterized by 25 constituents, containing 44.5% linalol, 10.9% borneol, and 9.8% terpinene-4-olo, out of 97.9% of the total components identified. In a study by Silva et al. [33], linalool was the predominant component of the *L. angustifolia* EO, accounting for 32.52% of the total of 28 components identified. Notably, the changes in the EO composition found in the literature were mainly attributed to the genetics of the plants and differences in their climatic, seasonal, and geographical conditions [34]. The results obtained in the present study agreed with those reported in the literature, with linalool, linalyl acetate, fenchone, eucalyptol, and borneol found to be the major LEO components [34,35,36,37,38].

### 3.2. In Vitro PBMC Viability after EO Treatment

The EOs used in vitro were tested for their toxic effects on sheep PBMCs by evaluating the cell viability within a concentration range from 3 to 10% (Figure 1a–c). The PBMC viability was significantly affected by all the EOs in the in vitro treatments (*p* < 0.001 for all EOs tested), showing a viability at least at the level of that of the negative control (CNS). In particular, the MEO treatment tested at the 10% concentration and stimulated with ConA, and its combination with LPS, tested at 5% and stimulated with ConA, resulted in a higher viability than the CNS, CS_ConALPS, CS_LPS, 3_ConALPS, and 5_ConALPS (Figure 1a). The REO treatment tested at 10% and stimulated with ConA and its combination with LPS showed the highest viability, as compared with all the controls (CNS, CS_ConA, CS_ConALPS, and CS_LPS, Figure 1b). Similarly, the LEO tested at 10% in the PBMCs stimulated with ConA and its combination with LPS showed a higher viability than all the in vitro treatments, except for that registered for the PBMCs treated with 7% LEO stimulated with ConA (Figure 1c).

The evaluation of the toxic effects of the EOs at different working concentrations can be considered a crucial step in the biological experiments conducted in vitro; thus, it is necessary to choose specific concentrations in compliance with the sample [10]. To the best of our knowledge, this is the first report in which the biological effects of MEO, REO, and LEO on sheep PBMCs were examined in terms of proliferation and cytokine production. The literature on LEO cytotoxicity places it in the category of “safe” oils [39]. Very few reports are available on its cytotoxicity. In the study by Prashar et al. [40], LEO was tested on different human skin cell lines, and, starting from 0.25% (*v*/*v*), a cytotoxic effect was registered with a similar pattern to that expressed when using linalool (contained in 35% lavender oil), suggesting that it is an active component of lavender oil [40]. Furthermore, REO tested at 1 mg/mL on hepatocellular carcinoma (Hep 3B & Hep G2) had no impact on cell viability [41]. This result was consistent with that reported by the European Medicines Agency on the clinical safety of REO [42], as well as with that by the European Food Safety Authority, which declared the safety of REO for dietary exposure [43]. Finally, a risk-assessment study on the cytotoxity of MEO found that the concentration of 100 μg/mL did not affect the viability of human epidermal keratinocyte (HaCaT) cells, which remained at approximately 90% [44]. Based on our data and according to the literature, the concentrations of all the EOs tested in the present experiment may be considered non-toxic for sheep-PBMC studies.

### 3.3. In Vitro PBMC Proliferative Response to Mitogen Stimulation and EO Treatments

Inflammation triggered by invading pathogens or endogenous signals, such as damaged cells, involves a series of complex and often simultaneous molecular, immunological, and physiological processes focused on the elimination of the initial cause of injury, the clearance of necrotic cells, and tissue repair [45]. For example, during an inflammatory response, an increase in blood leukocyte influx, vascular and intracellular cell-adhesion molecule (VCAM and ICAM) expression, the upregulation of the enzyme activity of oxygenase, peroxidase, and nitric oxide synthase, and the shift in metabolism of arachidonic acid with the release of pro-inflammatory cytokines are observed [46,47,48,49]. In the present paper, the effects of the EO treatments on the blastogenic responses of sheep cells when using a combination of LPS and ConA mitogens or ConA only were verified. The proliferation of PBMCs was significantly affected by the in vitro EO treatments (*p* = 0.008 for MEO, *p* = 0.005 for REO, and *p* = 0.0002 for LEO, respectively, Figure 2). The PBMCs treated with MEO at 10% and stimulated with ConA and LPS had greater proliferation than the positive control (CS ConA_LPS, Figure 2a). By contrast, the proliferation of PBMCs treated with REO at 5% and stimulated with ConA and LPS registered a greater proliferation than the treatments at 7% and 10%, which were at the same level as the negative control (CNS, Figure 2b). A greater proliferative response was displayed in the PBMCs treated with REO at 5% stimulated with ConA and LPS than in those stimulated only with ConA, which was probably due to the different responses of cells to mitogen. The proliferation of PBMCs treated with 10% LEO, and stimulated with ConA and LPS mitogens, was greater than those all the in vitro treatments, except for the proliferation of 10% stimulated with ConA only (Figure 2c).

When planning an in vitro study, it must be considered whether a particular compound or condition causes optimal proliferation [13]. Mitogens are compounds originating in plants (i.e., Con A) or microorganisms (i.e., LPS), which can be employed to test immunocompetence in animals in both in vitro and in vivo experiments. They activate DNA synthesis and the division of large leucocyte populations. In particular, mitogen can be classified into T-cell mitogens (i.e., Phorbol 12-myristate 13-acetate (PMA), ionomycin, A23187, Phytohemagglutinin, ConA, and others), B-cell mitogens (i.e., anti-IgM Ab, LPS, 8-mercaptoguanosine, protein kinase C activators, and others), and polyspecific mitogens, for both T- and B-cell proliferation, like Pokeweed Mitogen (PWM) [12]. Therefore, we decided to test two different types of blastogenic stimulators, the combination of ConA and LPS and ConA only, considering the lack of knowledge on this argument and to cover the possible absence of response to mitogens, which can cause a marked inhibition of the lymphocyte proliferation rate [13], and could mask the real effect of in vitro tests.

No studies were available on the blastogenic responses of EOs in sheep in an in vitro model of inflammation. Recent research mainly reported the anti-proliferative effects of linalool, one of the main chemical active compounds in LEO [50], *L. angustifolia* [51], *R. officinalis* [52], and *M.* x *piperita* Eos [53], on different tumor-derived cell lines. Very few findings reported the effects of EOs on peripheral blood leukocytes’ proliferation in humans. A previous study, conducted on human PBMC proliferation induced by PHA, found that the EO of *M.* x *piperita* L. var. RAC 541 had a stimulatory effect on proliferation by affecting cytokine production in different way, according to a specific cultivar [54]. Conversely, an experiment focused on the evaluation of the immunomodulatory effect of *Nigella sativa* EO (NSEO) on human PBMCs reported an antiproliferative effect on CD4^+^ and CD8^+^ T cells when NSEO was tested at the lowest (1:10, 1:50, and 1:100) dilutions [55]. Moreover, a reduction in the proliferation rate of peripheral blood leukocytes without affecting their capacity to secrete anti-inflammatory cytokines was demonstrated by using *Melaleuca alternifolia* EO [56]. In the present experiment, none of the concentrations of EOs tested exerted any antiproliferative effects on the PBMCs; on the contrary, the highest concentrations of MEO and REO and the 5% LEO increased the proliferative response to the mitogens ConA and LPS. The stimulatory action of EOs on PBMCs could be particularly desirable for treating immunocompromised animals, such as those exposed to physiological [57] and/or environmental stressors [58].

### 3.4. Cytokine Profiles in PBMC Supernatant

Cytokine secretion is triggered by stimuli like bacterial endotoxin or lipopolysaccharide (LPS), as well as many other pathogen components that similarly activate Toll-like receptors [59,60]. These receptors trigger a cascade of intracellular signals, involving the activation of nuclear factor kappa B (NFκB) and resulting in the early transcription of cytokines such as IL-1, IL-6, and tumour necrosis factor-α (TNF-α) [61]. The immunostimulatory effects of EOs are mainly related to their interaction with the signaling cytokines, the regulatory transcription factors, and the expression of pro-inflammatory genes [9]. Previous studies hypothesized that the mechanisms of action of EOs on immune cells can be both indirect and direct via several mechanisms, such as hyperemia, which accelerates the recruitment of leukocytes and promotes anti-inflammatory reactions (citrals, citronellal, and cuminal in external use), or the inhibition of the synthesis and secretion of inflammatory mediators (histamines, pro-inflammatory cytokines, prostaglandins, leukotrienes, nitric oxide, and free radicals), representing different levels of anti-inflammatory activity [62,63].

It is worth noting that the molecular basis of the immunomodulatory activities of EOs has not been deeply investigated and could change, depending on the specific EO used [10]. In the present paper, the secretion of IL-6 was not affected by the in vitro EO treatments (*p* = 0.232 for MEO, *p* = 0.492 for REO, and *p* = 0.524 for LEOs, respectively, Figure 3a). The level of IL-10 was higher in the lowest concentrations of both the REO and the LEO treatment, even though it displayed higher IL-10 values than the positive control (ConA_LPS, Figure 3b). Conversely, the level of IL-8 secretion in the PBMC supernatant was influenced differently in relation to the type of EO in vitro treatment. The MEO treatment significantly affected the level of IL-8 (*p* < 0.001, Figure 3c), as demonstrated by the lower levels registered in all the tested concentrations compared with both CS_ConALPS and CNS. The REO treatment at 7% showed a lower IL-8 secretion than the NC (*p* = 0.0018). Regarding the LEO treatment, all the tested concentrations had lower levels of IL-8 than the CNS (*p* = 0.0013).

The ability of LEO and its constituents to interfere with the nuclear factor kappa-light-chain-enhancer of activated B cells (NF-κB), and p38-mitogen-activated protein kinase (MAPK)-signaling immunological pathways, as well as cytokine secretion, was demonstrated [64,65]. Indeed, the main components of LEO, including (+)-α-pinene, (–)-linalool, and (+)-limonene, decreased the secretion of interleukin-2 (IL-2) and increased the IL-10/IL-2 ratio in mouse primary splenocytes, indicating their potential role in the activation of T-helper 2 (Th2) responses [65], which promotes the humoral immune response [66]. Furthermore, (– linalool was found to be able to attenuate the production of LPS-induced TNF-α and IL-6, both in RAW 264.7 macrophages, and in mice [64,65]. In the present paper, both LEO and REO treatments resulted, concomitantly, in a higher level of IL-10 and a lower level of IL-8 secreted by the PBMCs, demonstrating their potential role in the activation of the Th2 response, driven by IL-10. Interleukin-8 (IL-8) is a chemokine family member, identified as neutrophil chemotactic polypeptide, in the conditioned media of LPS-stimulated peripheral blood monocytes [67]. The regulation of IL-8 production or the inhibition of its action were investigated during acute inflammation with a therapeutic purpose [67]. Based on this evidence, the reduced secretion of IL-8 by PBMCs in the presence of EOs underlined the potential immunoregulatory role of EOs in both physiological and pathological conditions, which needs to be further explored. Indeed, results from an in vitro study attributed the anti-inflammatory activity of eight EOs to their inhibitory effect on NF-κB activation, with the downregulation of IL-6, IL-1β, TNF-α, and COX-2 mRNA expression in macrophages [10]. Furthermore, IL-10 is known as an anti-inflammatory cytokine that inhibits the production of a wide range of cytokines, including IL-1, IL-6, IL-8, and TNF-α, in various cell types, by causing the selective inhibition of NF-κB [68]. Based on this assumption, the higher level of IL-10 found in the PBMCs treated with EOs could help to justify the lower level of IL-8 registered. However, EO supplementation in piglets resulted in a potentiated immune response by improving the lymphocyte proliferation rate, phagocytosis rate, and immunoglobulin (Ig) G, IgA, IgM, C3, and C4 production [69,70], as well as by decreasing the major pro-inflammatory (TNF-α, and IL-6), and anti-inflammatory (IL-10) cytokines [71]. These last statements confirm that EO activities are complex, cannot be considered only one-sided (anti-inflammatory or pro-inflammatory) [9], and might be able to modify the immune response in a context-specific way. In the present study, the cytokine profile mainly depended on the type of EO tested and could affect both the anti- and the pro-inflammatory-cytokine pattern.

## 4. Conclusions

This is the first study in which the effects of three EOs on sheep PBMC proliferation and cytokine production were evaluated. The data from the present research demonstrate that *R. officinalis* L., *M.* x *piperita* L., and *L. angustifolia* L. EOs can have a stimulatory effect on PBMCs’ blastogenic responses to both LPS and ConA mitogens. Furthermore, the immunomodulatory activity of the EOs tested was ascertained by showing a cytokine pattern in favor of IL-10 anti-inflammatory cytokines, with a concomitant lower level of the pro-inflammatory IL-8.

The present study encourages the use of EOs as feed additives to improve animals’ immune competence in the current context of more sustainable livestock production and the “One Health” vision. Furthermore, the stimulatory action exerted by EOs on PBMC proliferation and cytokine secretion, could be an appropriate solution to treat immunocompromised animals involved in livestock production.

## Figures and Tables

**Figure 1 vetsci-11-00157-f001:**
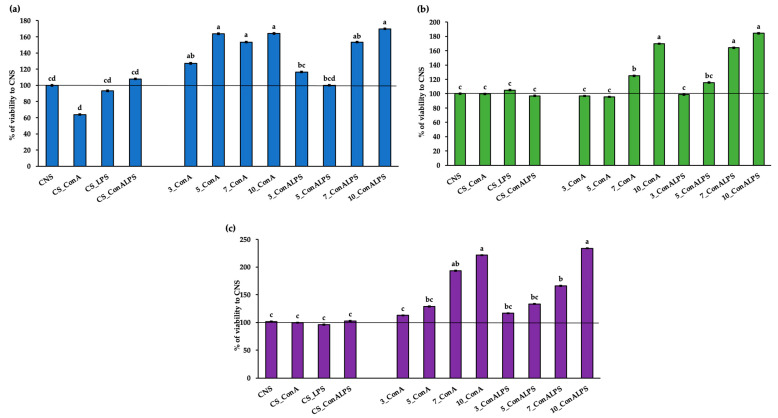
(**a**) PBMC-viability percentage after in vitro treatment with incremental concentrations (3, 5, 7, and 10%) of *Mentha* x *piperita* L. EO (MEO) stimulated with concanavalin A (ConA) and liposaccharide (LPS) and concanavalin A only (ConA); (**b**) PBMC-viability percentage after in vitro treatment with incremental concentrations (3, 5, 7, and 10%) of *Rosmarinus officinalis* L. EO (REO) stimulated with concanavalin A (ConA) and liposaccharide (LPS) and concanavalin A only (ConA); (**c**) PBMC-viability percentage after in vitro treatment with incremental concentrations (3, 5, 7, and 10%) of *Lavandula angustifolia* L. EO (LEO) stimulated with concanavalin A (ConA) and liposaccharide (LPS) and concanavalin A only (ConA). Controls were represented by CNS (unstimulated PBMC), CS_ConA (PBMCs stimulated with ConA), CS_ConALPS (PBMCs stimulated with ConA and LPS), CS_LPS (PBMCs stimulated with LPS). Treatments with differing superscripts (a, b, c, d) differ (*p* < 0.05).

**Figure 2 vetsci-11-00157-f002:**
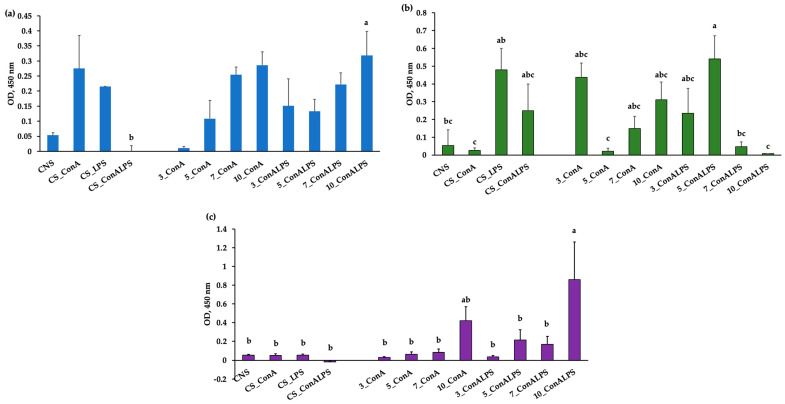
(**a**) PBMC proliferative response measured with optical density (OD) at 450 nm after in vitro treatment with incremental concentrations (3, 5, 7, and 10%) of *Mentha* x *piperita* L. EO (MEO) stimulated with concanavalin A (ConA) and liposaccharide (LPS) and concanavalin A only (ConA); (**b**) PBMC proliferative response measured with optical density (OD) at 450 nm after in vitro treatment with incremental concentrations (3, 5, 7, and 10%) of *Rosmarinus officinalis* L. EO (REO) stimulated with concanavalin A (ConA) and liposaccharide (LPS) and concanavalin A only (ConA); (**c**) PBMC proliferative response measured with optical density (OD) at 450 nm after in vitro treatment with incremental concentrations (3, 5, 7, and 10%) of *Lavandula angustifolia* L. EO (LEO) stimulated with concanavalin A (ConA) and liposaccharide (LPS) and concanavalin A only (Con)A. Controls were represented by CNS (unstimulated PBMC), CS_ConA (PBMC stimulated with ConA), CS_ConALPS (PBMC stimulated with Con and LPS), CS_LPS (PBMC stimulated with LPS). Treatments with differing superscripts (a, b, c) differ (*p* < 0.05).

**Figure 3 vetsci-11-00157-f003:**
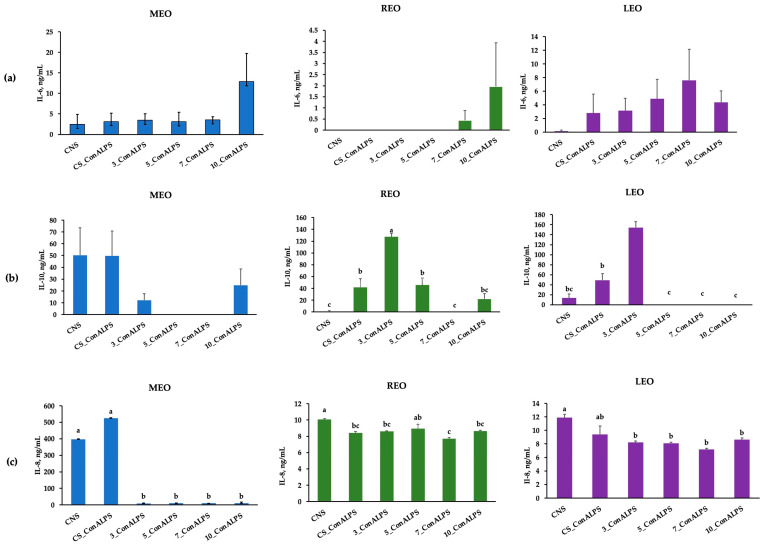
Secretion of (**a**) IL-6, (**b**) IL-10, and (**c**) IL-8 in PBMCs after in vitro treatment with incremental concentrations (3, 5, 7, and 10%) of EOs and in presence of the combination of concanavalin A (ConA) and liposaccharide (LPS) stimuli. Controls were represented by CNS (unstimulated PBMC), and CS_ConALPS (PBMC stimulated with ConA and LPS). In blue, data on *Mentha* x *piperita* L. EO (MEO), in green, data on *Rosmarinus officinalis* L.(REO), and in purple, data on *Lavandula angustifolia* L. (LEO) cytokine production. Treatments with differing superscripts (a, b, c) differ (*p* < 0.05).

**Table 1 vetsci-11-00157-t001:** Relative percentage compositions of *Rosmarinus officinalis* L., *Mentha* x *piperita* L., and *Lavandula angustifolia* L. essential oils used for the in vitro sheep PBMC treatments.

No.	*Mentha* x *piperita* L.	Percentage (%)	No.	*Rosmarinus* *officinalis* L.	Percentage (%)	No.	*Lavandula* *angustifolia* L.	Percentage (%)
1	**α-Pinene**	**1.6**	1	Tricyclene	0.4	1	α-Tujene	0.2
2	**Sabinene**	**1.5**	2	α-Thujene	0.3	2	α-Pinene	0.7
3	**β-Pinene**	**2.7**	3	**α-Pinene**	**20.0**	3	Camfene	0.4
4	**Myrcene**	**4.2**	4	**Camphene**	**10.8**	4	Sabinene	0.3
5	3-Octanol	0.3	5	**β-Pinene**	**4.9**	5	β-Pinene	0.6
6	**Limonene**	**11.8**	6	**3-Octanone**	**1.2**	6	1-octen-3-ol	0.6
7	**1,8-Cineole**	**17.4**	7	**β-Myrcene**	**5.1**	7	3-octanone	0.2
8	Linalol	0.2	8	**α-Phellandrene**	**2.4**	8	**β-Myrcene**	**1.0**
9	α-Terpineol	0.3	9	Δ3-Carene	0.1	9	α-Phellandrene	0.1
10	Piperitone oxide	0.4	10	α-Terpinene	1.0	10	Δ3-Carene	0.2
11	Carvone	0.2	11	**p-Cymene**	**1.1**	11	α-Terpinene	0.1
12	**Pipertitenone oxide**	**51.25**	12	**1,8-Cineole**	**25.2**	12	**p-Cymene**	**4.7**
13	**β-Caryophyllene**	**3.3**	13	β-cis-Ocimene	0.1	13	**1,8-Cineol**	**11**
14	Germacrene D	0.6	14	**γ-Terpinene**	**1.8**	14	**Trans Ocimene**	**3.3**
15	Bicyclogermacrene	0.1	15	Terpinalene	1.0	15	**Cis Ocimene**	**1.3**
16	Caryophyllene oxide	0.8	16	Linalol	0.7	16	γ-Terpinene	0.4
			17	**Camphor**	**16.2**	17	Terpinolene	0.4
			18	**Borneol**	**1.7**	18	**Linalol**	**44.5**
			19	Terpinen-4-olo	0.4	19	**Canfor**	**3.6**
			20	α-Terpineol	0.7	20	**Borneol**	**10.9**
			21	**Verbenone**	**1.4**	21	**Terpinen-4-ol**	**9.8**
			22	Bornyl acetate	0.7	22	α-Terpineol	0.9
			23	**β-Caryophyllene**	**1.6**	23	**Lavandulyl** **Acetate**	**1.3**
			24	α-Humulene	0.4	24	Geranyl Acetate	0.7
						25	β-Farnesene	0.7
	% Total identified	**96.65**		% Total identified	**99.2**		% Total identified	**97.9**

Compounds were numbered based on the order of retention times. Highlighted compounds are more than 1%.

## Data Availability

All data can be provided by the corresponding author upon reasonable request.

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
