# Peer review of "Immunomodulatory Role of Rosmarinus officinalis L., Mentha x piperita L., and Lavandula angustifolia L. Essential Oils in Sheep Peripheral Blood Mononuclear Cells"

_vetsci, 2024, doi:10.3390/vetsci11040157_

Round 1
Reviewer 1 Report
Comments and Suggestions for Authors
The paper is interesting and innovaitve especially for the future use of essential oils as natural treatment in the farm animals. The bibliography is relevant and sufficient. I suggest to put the phrase "The animal study protocol was approved by the Ethics 416 Committee of the University of Foggia (protocol number 0002302). " also in line 99 in MM Animals and experimental treatments.
The figures 1 and 2 are not clear, I suggest to modify to underline the significant differences.
Moreover in all paper, particularly in the abstract and in the results, the eventually significant results are not clear i suggest a revision of these paragraphs.
Author Response
The paper is interesting and innovaitve especially for the future use of essential oils as natural treatment in the farm animals. The bibliography is relevant and sufficient. I suggest to put the phrase "The animal study protocol was approved by the Ethics 416 Committee of the University of Foggia (protocol number 0002302). " also in line 99 in MM Animals and experimental treatments.
AU: Thanks for the positive comments, the suggestion has been implemented in the text.
The figures 1 and 2 are not clear, I suggest to modify to underline the significant differences.
AU: The figures have been changed, accordingly.
Moreover in all paper, particularly in the abstract and in the results, the eventually significant results are not clear i suggest a revision of these paragraphs.
AU: The paragraphs have been modified, accordingly.
Reviewer 2 Report
Comments and Suggestions for Authors
I think the study is original and interesting. I welcome the emphasis on the possibility of using essential oils in animal husbandry in order to limit the use of antibiotics and the elimination of greenhouse gases.
I reported some suggestions in the attached file. In general, I suggest an English editing of the manuscript in order to enhance its reading and comprehension.

English editing highly reccommended.
Author Response
Abstract
Consider rewriting the abstract. I suggest describing the tests differently, specifying which were the control tests(mentioned in line 29) and summarizing the results obtained highlighting only the most significant ones.
AU: We thank the reviewers’ for all his suggestions. We have modified the abstract accordingly.
Lines 23-25. Consider revising the sentence: ‘Peripheral blood mononuclear cells (PBMCs) were treated with incremental concentrations (3, 5, 7, and 10%) of each EOs, in presence of Concanavalin A (ConA) alone or in combination with lipopolysaccharide
AU: The sentence has been modified, accordingly. However, in order to clarify the controls used in the in vitro test, a slight modification has been applied.
Line 27. Consider to mode this sentence after line 22.
AU: The abstract has been changed based on the reviewer consideration.
Lines 28-33. Abbreviations MEO, LEO, REO
AU: The abbreviations were explained in the first line, however, to meet the reviewer’s suggestion a modification has been added.
Lines 28-29. Consider a synonym for “challenged.”
AU: The term “challenge” was previously utilized in many papers such as “Lipopolysaccharide challenge: immunological effects and safety in humans. Expert review of clinical immunology, 11(3), 409-418.” to identify the activation of the inflammatory response after a stimulus among which the LPS one. However, to meet the reviewer’s request the term has changed with “stimulated”.
Lines 29. You didn’t speak about control tests previously in the text.
AU: The specification has been added early in the text, accordingly.
Introduction
May you add some information about peripheral blood mononuclear cells (PBMCs) viability, proliferation, and cytokines secretion
AU: Some information on the cytokines have been added. On the contrary, the evaluation of the PBMCs proliferation was previously underlined (L63-63) as an in vitro evaluation of animals’ immune competence.
Lines 69-70 Consider revising the sentence. in order to optimize the lymphocyte proliferation rate in both in vivo and in vitro trials”.
AU: The sentence has been revised, accordingly.
Lines 71-74. Consider revising the sentence. “In this study, we investigated the effects of Mentha x piperita L., Rosmarinus officinalisL., and Lavandula angustifolia L. oils (EOs) on sheep peripheral blood monocular cells (PBMCs) viability, proliferation, and cytokines secretion, stimulated by ConA and the combination of ConA and LPS.
AU: The sentence has been revised, accordingly.
Line 77. “…identified as immune biomarkers”. Consider deleting this statement or explaining it in more detail.
AU: The statement has been deleted.
Result and Discussion
Consider using a synonym for “challenged” throughout the text.
AU: Please see comments at Lines 28-29.
Lines 183-185. Consider to rewriting this sentence.
AU: The sentence has been modified, accordingly.
Figures. I think the figures are very useful in presenting the results. Perhaps the written part can be further summarized by highlighting only the main aspects of the results.
AU: Thanks for the comments, the written part has been summarized by highlighting only the main aspects of the results as well as possible.
Reviewer 3 Report
Comments and Suggestions for Authors
A very interesting manuscript describing the in vitro immunomodulatory effects of essential oils on sheep PBMC. The results and discussion section was well-written and informative. It will be very interesting if the results will be similar in an in vivo study and if milk sheep will find diets containing essential oils palatable and if their milk will change in flavor.
L86: were herbicides used for weed control? How may this have effected essential oil components?
Section 2.2 Animals and experimental treatments: how many animals were used? What were their ages?
Sections 2.3-6: How many replicates were performed for each sample? Where experiments repeated on different dates? Were standards included on each plate? What types of plates were used? What volumes and cell numbers were included in the wells? Were the plates centrifuged prior to removing supernatants?
Table 1: how were the components numbered/ordered?
Figures need to be larger - it is difficult to read the treatment names and superscripts. A statement such as "treatments with differing superscripts differ" should be included in each figure legend.
Comments on the Quality of English Language
There are some problems with usage that can be improved, but the meaning of the sentences were understandable.
Author Response
A very interesting manuscript describing the in vitro immunomodulatory effects of essential oils on sheep PBMC. The results and discussion section was well-written and informative. It will be very interesting if the results will be similar in an in vivo study and if milk sheep will find diets containing essential oils palatable and if their milk will change in flavor.
L86: were herbicides used for weed control? How may this have effected essential oil components?
AU: Mechanical and manual weeding was carried out. No herbicides were used, so they did not affect upon composition of the essential oils.
Section 2.2 Animals and experimental treatments: how many animals were used? What were their ages?
AU: The animals were involved in the study were 5 of about 5 years old; however, in this experiment animals were considered as blood donors to isolate PBMC to test the EO in vitro effect; as a consequence, the number of animals were defined according to the 3R strategy in animal research.
Sections 2.3-6: How many replicates were performed for each sample? Where experiments repeated on different dates? Were standards included on each plate? What types of plates were used? What volumes and cell numbers were included in the wells? Were the plates centrifuged prior to removing supernatants?
AU: Regarding to both BrdU and XTT assays were run into quadruplicates (L138), moreover, the ELISA for ILs quantification was performed into duplicates. The experiment was repeated two times and on each plate were added the standards for each cytokine as reported at the L 164-166. The plates used were 96 U-bottom plate and the cell numbers seeded was 2*10^6 into 100 microliters as already reported in the line 138 for BrdU assay, and 1 x 105 cells/ml for XTT assay. Before the supernatant collection the plates were centrifuge to avoid collecting cells which were used for the evaluation of proliferation and viability tests. A brief specification of all requested suggestion has been added to clarify the methods, even if the citation Ciliberti et al. (2017) contained all the methods in an extensive form.
Table 1: how were the components numbered/ordered?
AU: The components are ordered based on the retention times. A specification has been added in the text.
Figures need to be larger - it is difficult to read the treatment names and superscripts. A statement such as "treatments with differing superscripts differ" should be included in each figure legend.
AU: The figures have been improved. The statement has been added, accordingly.
Round 2
Reviewer 2 Report
Comments and Suggestions for Authors
I do not have further suggestions. I thank the Authors for their work.
Comments on the Quality of English LanguageI think the English language is fine. Minor revisions may be required.